# Counteracting suppression of CFTR and voltage-gated K⁺ channels by a bacterial pathogenic factor with the natural product tannic acid

**Yajamana Ramu, Yanping Xu[†], Hyeon-Gyu Shin, Zhe Lu***

Department of Physiology, Perelman School of Medicine, Howard Hughes Medical Institute, University of Pennsylvania, Philadelphia, United States

**Abstract** Mutations in the cystic fibrosis transmembrane conductance regulator (CFTR) cause recurring bacterial infection in CF patients' lungs. However, the severity of CF lung disease correlates poorly with genotype. Antibiotic treatment helps dramatically prolong patients' life. The lung disease generally determines prognosis and causes most morbidity and mortality; early control of infections is thus critical. *Staphylococcus aureus* is a main cause of early infection in CF lungs. It secretes sphingomyelinase (SMase) C that can suppress CFTR activity. SMase C also inhibits voltage-gated K⁺ channels in lymphocytes; inhibition of these channels causes immunosuppression. SMase C's pathogenicity is further illustrated by the demonstration that once *Bacillus anthracis* is engineered to express high levels of SMase C, the resulting mutant can evade the host immunity elicited by a live vaccine because additional pathogenic mechanisms are created. By screening a chemical library, we find that the natural product tannic acid is an SMase C antidote.

**\*For correspondence:** zhelu@ mail.med.upenn.edu

[†]Deceased

**Reviewing editor**: Richard Aldrich, The University of Texas at Austin, United States

## Introduction

The cystic fibrosis transmembrane conductance regulator (CFTR) Cl⁻ channel is activated when its regulatory (R) domain is phosphorylated by cyclic AMP-dependent protein kinase A (PKA) (*Tabcharani et al., 1991*). Mutations of CFTR cause CF disease that involves multiple organs (*Knowles et al., 1983*; *Kerem et al., 1989*; *Riordan et al., 1989*; *Rommens et al., 1989*). In the lungs of CF patients, defective CFTR leads to production of thick mucus that obstructs airways and thus predisposes the patients to recurring bacterial infection. Although CF disease originates from genetically defective CFTR protein, the severity of CF lung disease is not well correlated with genotype. The CF lung pathology is not fundamentally different from those of many other types of chronic pulmonary infectious and inflammatory diseases. Aggressive antibiotic treatment and supportive measures have prolonged the median life span of patients from 5 to 37 years (*The Cystic Fibrosis Foundation, 2004*). These observations underscore the profound impact of infections on progression and severity of CF lung disease. Given that the lung disease generally determines CF patients' prognosis and causes ~90% of their morbidity and mortality, it is extremely important to effectively control or mitigate lung infections in the early stages of CF disease.

*Staphylococcus aureus* is a main cause of early infection in the lungs of CF patients (*The Cystic Fibrosis Foundation, 2008*). Most *S. aureus* strains, including methicillin-resistant *S. aureus* (MRSA), secrete the pathogenic factor sphingomyelinase (SMase) C, which cleaves sphingomyelin into phosphocholine and ceramide (*Doery et al., 1963*) (upper panel, *Figure 1A*). Our group has previously discovered that SMase C hydrolysis of sphingomyelin surrounding CFTR protein profoundly suppresses CFTR activity (*Ramu et al., 2007*). Thus, bacteria can further diminish critical residual CFTR

**eLife digest** Cystic fibrosis is an inherited disease that mainly affects the lungs and the intestine. It is caused by defective copies of a protein called CFTR, which normally allows chloride ions to flow in and out of cells through the cell membrane. The activity of the CFTR protein is important for transferring fluid in and out of cells, and the dysfunctional CTFR protein causes the cells lining the lungs and the intestine to produce thick mucus. This leads to problems with breathing and absorbing nutrients, and makes the lungs of people with cystic fibrosis more susceptible to infections.

Recurring lung infections aggravate the symptoms of cystic fibrosis and worsen the predicted outcome for sufferers. A common skin bacterium, called *Staphylococcus aureus*, is one of the first to colonize the lungs of young cystic fibrosis patients. This pathogen releases an enzyme that can further reduce any residual activity of the defective CTFR protein. The enzyme—called sphingomyelinase C (or SMase C for short)—also interferes with the function of another protein that allows potassium ions to flow out of immune cells. This effect in turn reduces the body's ability to fight the infection.

Now Ramu et al. have found, by using cells grown in a laboratory, that the enzyme released by the bacteria would remain active long after the bacteria had been killed. This finding suggests that a combination therapy of antibiotics that kill the bacteria and a drug that inhibits the enzyme function may greatly improve treatments for cystic fibrosis.

Ramu et al. then tested a collection of over 2000 edible naturally-occurring chemicals and drugs, which are already approved for use in humans, to see if any could counteract the activity of the enzyme. A single natural chemical called tannic acid was shown to prevent both the negative effects on CTFR and those on the potassium channel protein.

Other pathogenic bacteria also produce an SMase C enzyme and Ramu et al. showed that tannic acid can also interfere with the anthrax bacterium's enzyme. This suggests that treatment with tannic acid may improve the outcome in a number of bacterial infections. Further experimental work is now needed to establish whether tannic acid can alleviate the symptoms of infectious disease in animal models.

activity in CF patients, worsening the negative impact of mutations. By inhibiting CFTR, SMase C-producing bacteria can also create a temporary condition, analogous to CFTR deficiency, in non-CF patients with lung infection. Additionally, compounding this already very serious problem, SMase C strongly inhibits the activity of Kv1.3 voltage-gated K⁺ (Kv) channels of lymphocytes (*Xu et al., 2008*). Inhibition of these channels is well known to cause immunosuppression (*Chandy et al., 2004*).

SMase C's pathogenicity is further illustrated by the disturbing characteristics reported for a genetically engineered *Bacillus anthracis* mutant. Natural *B. anthracis*, against which the live STI-1 (Sanitary Technical Institute, USSR) vaccine provides effective protection, produces little SMase C due to a 'defective' regulatory gene. However, once *B. anthracis* is engineered to express high levels of SMase C, the resulting mutant not only remains lethal but also evades the host immunity elicited by the live vaccine because additional pathogenic mechanisms are generated (*Pomerantsev et al., 1997*). Thus, it is important to find effective means to counteract the SMase C action.

## Results and discussion

Experimentally, the CFTR channels are often activated by boosting the concentration of intracellular cAMP with a combination of the adenylate cyclase activator forskolin and the phosphodiesterase inhibitor isobutylmethylxanthine (IBMX) (*Csanady et al., 2000*). *Figure 1B* shows current of CFTR heterologously expressed in a *Xenopus* oocyte bathed in a forskolin- and IBMX-containing solution. Following exposure to extracellular SMase C from *S. aureus*, the CFTR current was markedly diminished in a time-dependent manner (*Figure 1C–E*). To investigate whether SMase C also suppresses native CFTR current, we tested the enzyme's effect on CFTR current in Calu-3 cells (a lung epithelial cell line commonly used in CFTR research) after activating their CFTR channels with forskolin and IBMX (*Figure 1F–H*). As in oocytes, SMase C suppressed CFTR current in Calu-3 cells (*Figure 1G–I*), a result confirming that this SMase C effect on CFTR activity is not unique to the oocyte heterologous expression system.

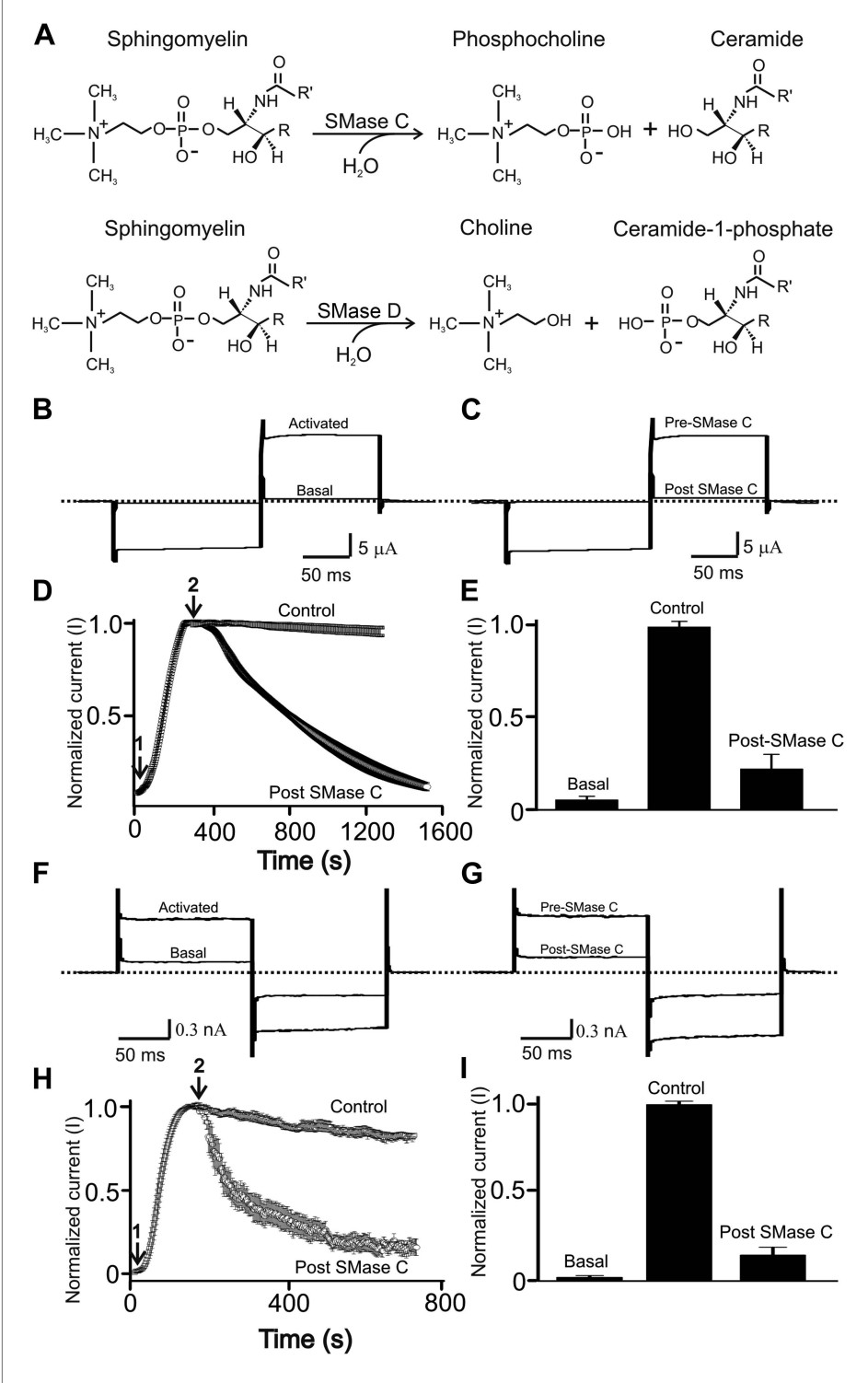

**Figure 1**. SMase C suppression of heretologously expressed and native CFTR currents. (**A**) Sphingomyelin hydrolysis reactions catalyzed by SMases C and D. (**B**) and (**C**) Currents of an oocyte injected with cRNA encoding CFTR before or after activation with 50 μM forskolin and 1 mM IBMX in the bath solution (**B**) and activated CFTR currents pre- or post-application of SMase C from *S.aureus* (SaSMase C: 0.4 ng/μl) (**C**), elicited by stepping membrane voltage from the −30 mV holding potential to −80 mV and then to 50 mV. (**D**) Time course of normalized CFTR currents at 50 mV where the arrows indicate addition of forskolin plus IBMX (arrow 1) or SaSMase C (arrow 2)

*Figure 1. Continued on next page*

*Figure 1. Continued*

(mean ± s.e.m, n = 6). (**E**) Normalized current amplitude before or after activation by forskolin plus IBMX and post-SMase C treatment (mean ± s.e.m, n = 6). (**F**) and (**G**) Native currents of a Calu-3 cell before or after activation with 50 µM forskolin and 1 mM IBMX in the bath solution (**F**) and activated CFTR currents before and after addition of $^{Sa}$SMase C (**G**), elicited by stepping membrane voltage from the 0 mV holding potential to −80 mV and then to 80 mV. (**H**) Time course of normalized CFTR currents at 80 mV where the arrows indicate addition of forskolin plus IBMX (arrow 1) or $^{Sa}$SMase C (0.5 ng/µl; arrow 2) (mean ± s.e.m, n = 7). (**I**) Normalized current amplitude before or after activation by forskolin plus IBMX and post-SMase C treatment (mean ± s.e.m, n = 7).

Individual strains of a given bacterium species may be expected to secrete different isoforms of SMase C. We compared three isoforms from *S. aureus* to learn their behaviors (**Figure 2A**; only isoform #1 was used elsewhere). All three recombinant isoforms suppressed CFTR current albeit with differing specific activity (**Figure 2B**). Two isoforms lost activity over 96 hr, and one retained full activity; all three isoforms exhibited measurable activity for at least 48 hr. It follows that secreted SMase C can remain active for a considerable time following death of the bacteria. Thus, while antibiotics remain critical in combating the infection, an effective means to counteract SMase C action would be important in limiting the negative impact of the bacterial infection on the host.

We previously found that at levels beyond that required to achieve maximal CFTR activation, PKA's catalytic subunit significantly lessens suppression of CFTR current by SMase C (**Ramu et al., 2007**). This observation suggests that SMase C modifying the head group chemistry of sphingomyelin molecules surrounding CFTR makes activation of CFTR more energetically difficult, and this SMase C action can be markedly overcome by 'over-phosphorylating' the channel's R-domain until all four sites are phosphorylated. This hypothesis implies that mutant CFTR channels with individual key phosphorylation sites ablated would suffer more pronounced suppression by SMase C at these 'supersaturating' levels of PKA's catalytic subunit. To test this prediction, we replaced each of the four phosphorylation sites, one at a time, with alanine and examined SMase C's effect on these single-mutant channels. As a control, we showed that in the presence of a supersaturating level of PKA's catalytic subunit, SMase C suppressed only 25% of the wild-type CFTR current (**Figure 2C**). In contrast and as predicted, under the same experimental condition, SMase C suppressed 40–65% of the mutants' currents. A second

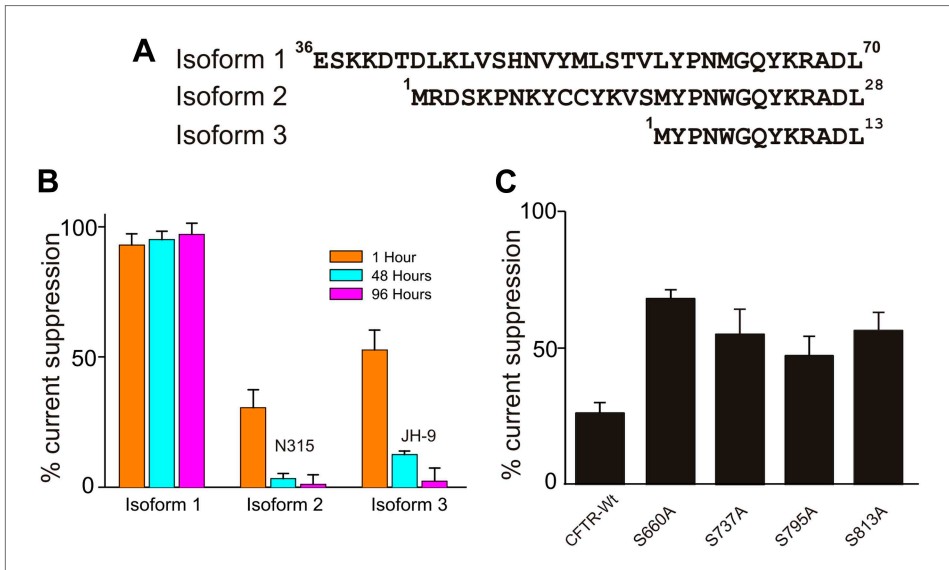

**Figure 2**. Effects of isoforms and mutations on $^{Sa}$SMase C suppression of CFTR current. (**A**) N-terminal sequences of three $^{Sa}$SMase C isoforms. (**B**) Percentage of $^{Sa}$SMase C suppression of CFTR current by individual isoforms at 1, 48 and 96 hr after final purification of SMase C (mean ± s.e.m, n = 5), where the final concentration was 0.4 ng/µl for isoform 1 and 40 ng/µl for isoforms 2 and 3. (**C**) Percentage of $^{Sa}$SMase C suppression of wild-type and mutant CFTR currents activated by expressing cRNA encoding the PKA catalytic subunit (mean ± s.e.m, n = 7–10).

corollary of our hypothesis is that a β adrenergic receptor agonist should alleviate the SMase C-caused suppression of CFTR current by boosting intracellular cAMP. In fact, β receptor agonists are routinely used as bronchodilators in patients suffering from pulmonary infectious and inflammatory diseases. Thus, application of β receptor agonists would not be contraindicated in cases of infection by SMase C-positive bacteria and likely be beneficial. However, given that a β receptor agonist unlikely causes overphosphorylation of the R domain, it may only modestly counteract the SMase C-dependent suppression of CFTR current.

To develop a clinically suitable drug from scratch is difficult and costly. One shortcut is to find novel medical benefits in already approved drugs or edible natural products. As such, we screened a chemical library (Spectrum 2000) consisting of approved drugs and natural products, each at 10 μM concentration. We employed a fluorescence-based assay (Amplex Red) involving a series of coupled enzymatic reactions. *Figure 3A* is a correlation plot of the Z scores obtained from two repeated screens. A positive result suggests an antagonizing effect of the tested compound on SMase C or on one or more of the coupled reactions. The 27 compounds with higher Z scores were then examined electrophysiologically for their antagonistic effects on SMase C (*Figure 3B*). We found that, of those 27 compounds at 10 μM, only tannic acid exhibited a clear SMase C-antagonizing effect (*Figure 3B,C*). Tannic acid was just as effective at concentrations down to 200 nM (*Figure 3D*). Furthermore, we found that it also antagonized SMase C of *B. anthracis* (*Figure 4A–C*).

Tannic acid is known to interact with many other proteins with apparent $K_d$ or $EC_{50}$ between 8 μM and 150 mM (*Gabbott, 2008*). Consequently, high concentrations of tannic acid (5–50 mM) are often used to fix cells in EM studies (*Hayat, 1981*; *Afzelius, 1992*; *Bozzola and Russell, 1992*). To our knowledge, a $EC_{50}$ range of 50–100 nM observed in our study is by far the lowest concentration range at which tannic acid is found to bind effectively to a protein and/or to produce a meaningful biological effect (*Figure 3F*). Although tannic acid has been reported to bind to the choline group of phosphatidylcholine (PC), it does not appear to antagonize SMase C activity by binding to PC and thereby preventing SMase C from accessing sphingomyelin because tannic acid binds to PC only at much higher concentrations than what is required to antagonize SMase C activity (*Kalina and Pease, 1977*; *Simon et al., 1994*). Furthermore, the $EC_{50}$ (~50 nM) of tannic acid for antagonizing the SMase C-catalyzed sphingomyelin hydrolysis in a biochemical reaction (involving no PC) is comparable to its $EC_{50}$ (~100 nM) for antagonizing SMase C suppression of CFTR activity in a biological membrane (*Figure 3F*).

To assess the relative specificity of tannic acid's antagonizing effect on SMase C, we tested tannic acid against bacterial SMase D. Like SMase C, SMase D specifically hydrolyzes sphingomyelin (bottom panel, *Figure 1A*). However, unlike SMase C, SMase D removes only the choline group rather than the entire phosphocholine head group from sphingomyelin, leaving ceramide-1-phosphate instead of ceramide behind in the membrane. Our group previously showed that SMase D also suppresses CFTR activity (*Ramu et al., 2007*). Here, we show that tannic acid at 300 nM slightly slowed down SMase D suppression of CFTR activity (*Figure 3E*). In contrast, as shown above, tannic acid at 200 nM essentially eliminated SMase C suppression of CFTR activity (*Figure 3D*). Thus, tannic acid antagonizes SMase C and SMase D with differing potency (*Figure 3F*).

In lymphocytes, Kv1.3 channels help maintain a sufficiently negative resting membrane potential to sustain an adequate driving force for $Ca^{2+}$ entry which acts as a key signal to activate lymphocytes (*DeCoursey et al., 1984*; *Matteson and Deutsch, 1984*; *Cahalan and Chandy, 2009*). Consequently, inhibition of Kv1.3 channels causes immunosuppression, a finding that has stimulated the development of inhibitors of Kv1.3 as effective immunosuppressants to treat autoimmune diseases (*Chandy et al., 2004*). On this backdrop, we previously showed that SMase C, by removing the phosphoryl head group of sphingomyelin molecules around Kv1.3 channels, profoundly diminishes these channels' activity (*Xu et al., 2008*). The underlying mechanism is that removal of the negatively charged phosphoryl group makes it energetically more difficult for positively charged voltage sensors to transition to the activated state. Here, we find that tannic acid can also effectively counteract suppression of Kv1.3 activity by SMase C (*Figure 4D–F*).

As illustrated above, we have shown that tannic acid is an antidote to SMase C. It is a readily available and inexpensive natural product, widely and abundantly used in the food industry. The US Food and Drug Administration terms it *Generally Recognized As Safe* (GRAS) (*FDA, 2013*). The use of tannic acid to treat various diseases was reported as early as 1850 (*Alison, 1850*; *Allegrini and Costantini, 2012*), for example in the treatment of human diarrhea. Recently reported clinical doses are 0.5–1 g (*Bian et al., 2009*), giving an estimated concentration in human body fluids of 8–16 μM, or 40–80 times

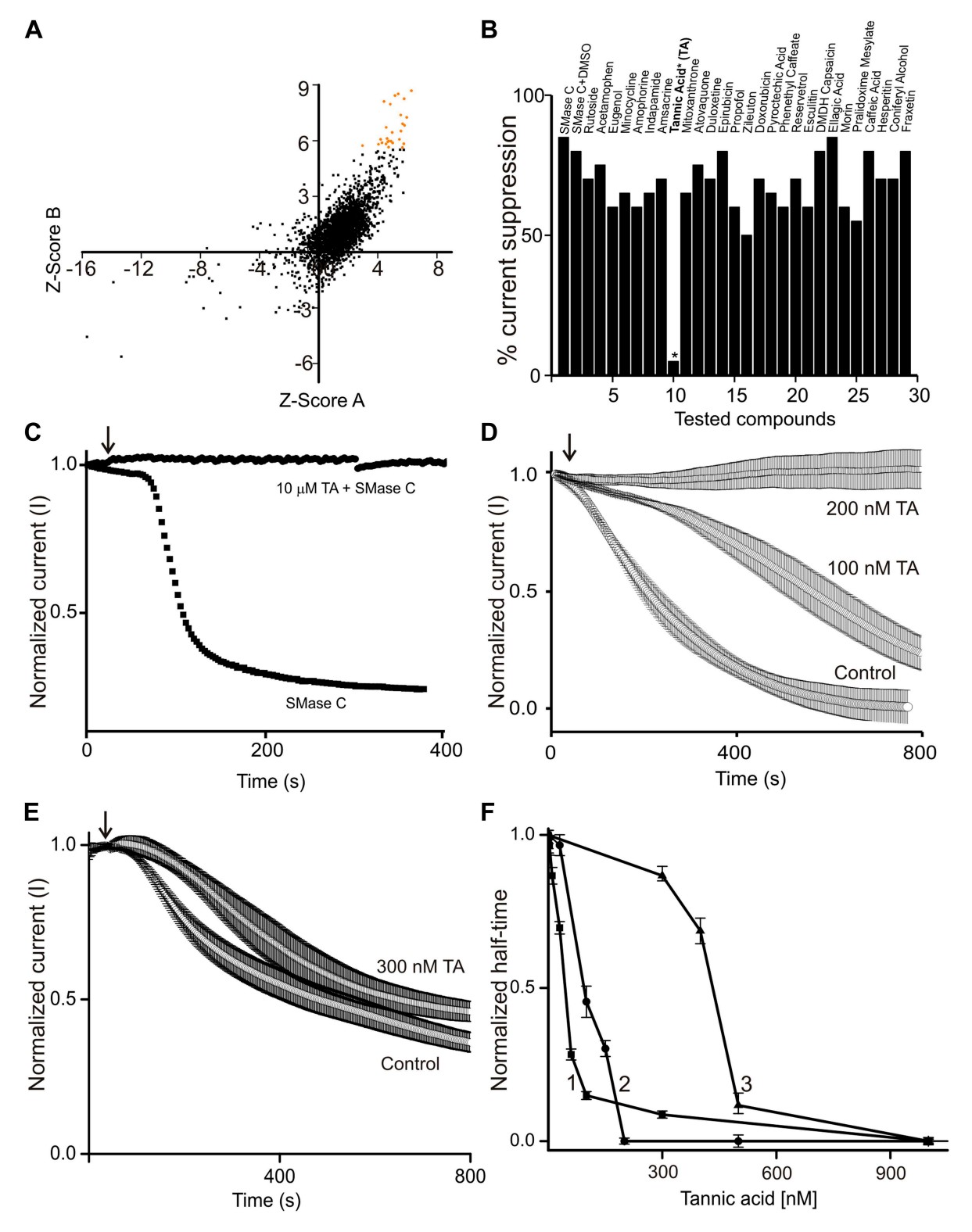

**Figure 3**. Tannic acid (TA) counteracts [Sa]SMase C and SMase D suppression of CFTR current. (**A**) Correlation plot of Z scores (raw scores minus the mean of the population, divided by the standard deviations of individual compounds) of individual compounds tested in two repeated screens; 27 compounds with higher Z scores are colored orange. (**B**) Percentage [Sa]SMase C suppression of CFTR currents in the presence of 27 tested compounds, each at 10 μM. (**C**) Time course of normalized CFTR currents at 50 mV without or with 10 μM tannic acid (TA) present in the bath; arrow indicates addition of

*Figure 3. Continued on next page*

*Figure 3. Continued*

[Sa]SMase C (0.6 ng/µl). (**D**) and (**E**) Time courses of normalized CFTR currents at 50 mV without and with 100 or 200 nM (**D**) or 300 nM (**E**) tannic acid present in the bath; arrow indicates addition of [Sa]SMase C (**D**) and SMase D (**E**) (mean ± s.e.m, n = 3–5). (**F**) Apparent activities of [Sa]SMase C (1 and 2) or SMase D (0.6 ng/µl) (3) plotted against the concentration of tannic acid, where the activities are expressed as half-time of the time course of sphingomyelin hydrolysis (1) or CFTR inhibition (2 and 3), which were obtained by the Amplex red assay (1) or by electrophysiological recordings (2 and 3). The half-time values were normalized to the corresponding ones obtained in the presence of relevant enzymes and the absence of tannic acid.

higher than required to antagonize SMase C action. We suggest that tannic acid be considered in attempts to protect against *B. anthracis* mutants engineered to over-express SMase C, in the unfortunate event of an outbreak, or to lessen the harm caused by natural SMase C-positive bacteria including MRSA, in both CF and non-CF patients.

# Materials and methods

## Molecular biology and electrophysiological recordings

The CFTR and PKA-C cDNAs were subcloned in the pGEMHE plasmid (*Liman et al., 1992*). Mutant CFTR cDNAs were obtained through PCR-based mutagenesis and confirmed by DNA sequencing. The cRNAs were synthesized with T3 or T7 polymerase using the corresponding linearized cDNAs as templates. Channel currents were recorded from whole oocytes previously injected with the corresponding cRNAs and stored at 18°C, using a two-electrode voltage clamp amplifier (OC-725C; Warner, Hamden, CT), filtered at 5 kHz and sampled at 50 kHz using a Digidata 1322 interfaced to a PC. The resistance of electrodes filled with 3 M KCl was 0.2–0.3 MΩ. Molecular Devices pClamp 8 software was used for amplifier control and data acquisition. Unless specified otherwise, the bath solution contained (in mM): 95 NaCl, 5 KCl, 0.3 CaCl$_2$, 1 MgCl$_2$ and 10 HEPES; pH was adjusted to 7.6 with NaOH. CFTR current from the Calu-3 cell line was recorded in the whole cell configuration with a patch-clamp amplifier (200B; Axopatch), filtered at 5 kHz and sampled at 50 kHz using a Digidata 1322 interfaced to a PC. Electrodes were fire polished (2–4 MΩ) and coated with beeswax. Capacitance and series resistance were electronically compensated. Molecular Devices pClamp 8 software was used for amplifier control and data acquisition. The bath solution contained (in mM) 145 NaCl, 5 KCl, 0.3 CaCl$_2$, 1 MgCl$_2$, 10 HEPES (pH 7.30 adjusted with NaOH) and the electrode solution contained (in mM) 140 KCl, 10 EGTA, 1 CaCl$_2$, 1 MgCl$_2$, 10 HEPES (pH 7.30 adjusted with KOH). SMases were manually added to the recording chamber to test their effects.

## Production of recombinant SMases

The cDNAs of SMase C were produced with PCR, primed with a pair of oligonucleotides corresponding to the 5′ or 3′ translated regions against the genomic DNA isolated from *B. anthracis* and *S. aureus* (isoform 1, accession number AAB32218; isoform 2, BAB43091 and isoform 3, ZP_01242591), respectively. To produce recombinant SMases C and D, *Escherichia coli* BL21 (DE3) cells were transformed with the respective cDNAs cloned into pET30 vector (Novagen, San Diego, CA), grown in LB medium to ~0.6 OD at 600 nm, and induced with 1 mM IPTG for 2 hr. The bacteria were harvested, resuspended, and sonicated. The resulting samples were loaded onto a cobalt affinity column and eluted by stepping the imidazole concentration from 50 to 500 mM (all SMase proteins contain N- and C-terminal His tags). The imidazole was later removed by dialysis.

## Quantification of SMase C inhibition with an Amplex red fluorescence-based assay

We screened the Spectrum 2000 compound library with Amplex red assay which is based on the following coupled reactions. First, sphingomyelinase hydrolyzes sphingomyelin, yielding ceramide and phosphocholine. Second, an alkaline phosphatase hydrolyses phosphocholine, yielding choline. Third, choline is oxidized by choline oxidase to betaine and H$_2$O$_2$. Finally, H$_2$O$_2$, in the presence of horseradish peroxidase, reacts with Amplex Red reagent in a 1:1 stoichiometry to generate the highly fluorescent product, resorufin. Resorufin has absorption and fluorescence emission maxima of approximately 571 nm and 585 nm, respectively. During the screen, individual wells of a PerkinElmer OptiPlateTM-384 HB black plate containing Spectrum 2320 compounds (each at 10 µM) dissolved in DMSO were added with 20 µl of a mixture containing sphingomyelin (100 µM), alkaline phosphatase

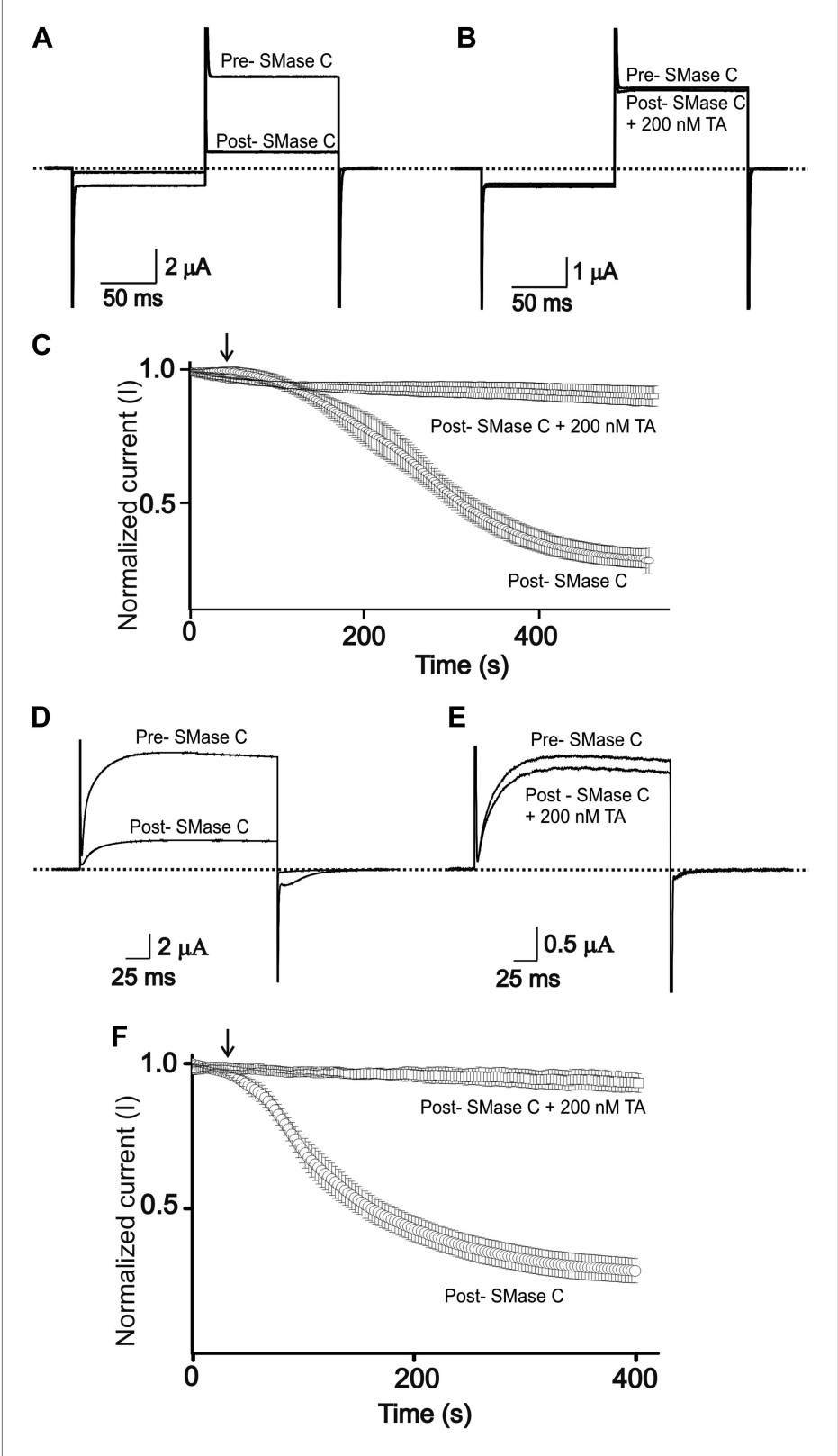

**Figure 4**. Tannic acid antagonizes suppression of CFTR and Kv1.3 currents by SMase C of *B.anthracis* ([Ba]SMase C). (**A**) and (**B**) Currents of an oocyte injected with cRNA encoding CFTR, which was activated as in *Figure 1*, before (pre) and after (post) addition of [Ba]SMase C (0.6 ng/μl) without (**A**) and with (**B**) 200 nM TA present. (**C**) Time course

*Figure 4. Continued on next page*

*Figure 4. Continued*

of normalized CFTR currents at 50 mV without or with 200 nM TA present in the bath; arrow indicates addition of <sup>Ba</sup>SMase C (mean ± s.e.m, n = 10). (**D**) and (**E**) Currents of an oocyte injected with cRNA encoding Kv1.3 before (pre) and after (post) addition of <sup>Ba</sup>SMase C without (**D**) and with (**E**) 200 nM TA, elicited by stepping membrane voltage from the −80 mV holding potential to 20 mV and then back to −80 mV. (**F**) Time course of normalized Kv1.3 currents at 20 mV without or with 200 nM TA in the bath; arrow indicates addition of <sup>Ba</sup>SMase C (mean ± s.e.m, n = 4).

(16 U/ml), choline oxidase (0.4 U/ml), horseradish peroxidase (4 U/ml), Amplex red (100 μM), and triton X (0.4%) (Amplex red sphingomyelinase assay kit A12220; Invitrogen, Grand Island, NY), using a Bio Tek microdispenser (Janus Automated Workstation, PerkinElmer, Waltham, MA). To start the enzymatic reaction, 20 μl of a solution containing <sup>Sa</sup>SMase (30 nM) was dispensed into individual wells. For chemical library screening, SMase C activity was quantified from the intensity ratio of fluorescence excited at 570 nm and detected at 590 nm in the presence and absence (DMSO only) of tested compounds whereas it was quantified by monitoring absorbance at 570 nm in the tannic acid dose–response study.

## Acknowledgements
We thank D Gadsby for CFTR cDNA in the pGEMHE vector, S Billington for SMase D cDNA, D Schultz and S Hodwadekar for advice and technical assistance, and P De Weer for critical review of the manuscript. Small molecule screening assay was performed in the Molecular Screening Facility of the Wistar Institute. Z Lu is an Investigator of the Howard Hughes Medical Institute.

## Additional information

### Funding

| Funder | Grant reference number | Author |
| --- | --- | --- |
| National Institute of General Medical Sciences | GM55560 | Yajamana Ramu, Yanping Xu |
| Howard Hughes Medical Institute | | Hyeon-Gyu Shin, Zhe Lu |

The funders had no role in study design, data collection and interpretation, or the decision to submit the work for publication.

### Author contributions
YR, Conception and design, Acquisition of data, Analysis and interpretation of data, Drafting or revising the article; YX, Conception and design, Acquisition of data, Analysis and interpretation of data; H-GS, Acquisition of data, Analysis and interpretation of data; ZL, Conception and design, Analysis and interpretation of data, Drafting or revising the article

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
