## [Decision Letter]

Thank you for sending your work entitled “Means to counteract suppression of CFTR and Kv1.3 channels by the bacterial pathogenic factor sphingomyelinase C” for consideration at *eLife*. Your article has been favorably evaluated by Vivek Malhotra (Senior editor) and 3 reviewers, one of whom is a member of our Board of Reviewing Editors.

The following individuals responsible for the peer review of your submission have agreed to reveal their identity: Richard Aldrich (Reviewing editor); Chris Miller and David Gadsby (peer reviewers).

The Reviewing editor and the other reviewers discussed their comments before we reached this decision, and the Reviewing editor has assembled the following comments to help you prepare a revised submission.

This interesting research extends the Lu lab's idiosyncratic approach to understanding, and ameliorating, the disease cystic fibrosis (CF). Mainstream CF research largely aims to correct defective folding and trafficking of the CFTR channel and to boost function of residual CFTR, so improving the flow of Cl- ions, and consequently water, across the critical epithelia of the lung and pancreatic and vas deferens ducts. Most CF morbidity and mortality results from bacterial colonization of dehydrated mucus in the lung and subsequent bacterial infections. The Lu lab has shown that snipping of the phosphocholine headgroup from sphingomyelin by secreted bacterial sphingomyelinase C (SMase C) diminishes Cl- current through adjacent CFTR channels as well as K+ current through lymphocyte Kv channels required for the immune response. Arguing that this two-pronged assault by bacterial SMase C in the lung contributes importantly to both CF and non-CF lung disease, the Lu lab report here their search for a small-molecule inhibitor of SMase C that included screening a library of 2,000 approved drugs and natural products. They found a promising candidate, tannic acid, active below 200 nM.

The work is characteristically novel, thorough, and convincingly written, and it warrants presentation to a broad audience. The final sentence of the paper lays down a challenge to the clinical CF community: “We suggest that tannic acid be considered in attempts....to lessen the harm caused by natural SMase C-positive bacteria including MRSA, in both CF and non-CF patients”. Prominent publication of this statement is important.

An additional important implication is that tannic acid could be used for treatment for engineered anthrax variants that are resistant to available vaccines.

The study might be faulted for the authors' explicit intent to eschew any biological studies of infection. But we consider this reasonable in light of their apparent purpose: to get the basic result known to the wider clinical world, so that proper biological studies can be carried out by those in a position to do them.

The following points need to be adequately addressed before the paper can be accepted for publication:

1) One glaring absence here is a proper biochemical tannic acid-inhibition curve for isolated SMase. That is far more pertinent than some of the other results reported here, such as ruling out PC as substrate.

2) The authors address the question of whether prevention of SMase C hydrolysis of sphingomyelin by tannic acid might result exclusively from tannic acid binding to the phosphocholine moiety of PC and then sterically impeding access of SMase C to nearby sphingomyelin. They apply PC-specific PLC to cleave phosphocholine from PC, and show that this application neither precludes SMase C diminution of Kv1.3 (as they also previously found for Shaker channels), nor prevents tannic acid from inhibiting that action of SMase. The problem is that the authors' interpretation of these observations requires that the PC-PLC removed the phosphocholine from PC in the oocyte membranes, and yet there is no evidence here (and little in [29]) that the PC-PLC was active. BSA would be expected to yield identical results. These negative results cry out for a positive control.

---

## [Author Response]

*1) One glaring absence here is a proper biochemical tannic acid-inhibition curve for isolated SMase. That is far more pertinent than some of the other results reported here, such as ruling out PC as substrate*.

We now include a biochemical tannic acid-inhibition curve for SMase C as well as functional tannic acid-inhibition curves for SMase C and SMase D in Figure 3.

*2) The authors address the question of whether prevention of SMase C hydrolysis of sphingomyelin by tannic acid might result exclusively from tannic acid binding to the phosphocholine moiety of PC and then sterically impeding access of SMase C to nearby sphingomyelin. They apply PC-specific PLC to cleave phosphocholine from PC, and show that this application neither precludes SMase C diminution of Kv1.3 (as they also previously found for Shaker channels), nor prevents tannic acid from inhibiting that action of SMase. The problem is that the authors' interpretation of these observations requires that the PC-PLC removed the phosphocholine from PC in the oocyte membranes, and yet there is no evidence here (and little in*
[29]*) that the PC-PLC was active. BSA would be expected to yield identical results. These negative results cry out for a positive control*.

It requires presentation of certain biological responses of oocytes to PC-PLC (which are not directly relevant to the present study) to clearly demonstrate that PC-PLC was active. We agree with the reviewers (major comment #1) that a biochemical tannic acid-inhibition curve for SMase C would rule out involvement of PC-PLC. Also, we now state in the discussion:

“Although tannic acid has been reported to bind to the choline group of phosphatidylcholine (PC), it does not appear to antagonize SMase C activity by binding to PC and thereby preventing SMase C from accessing sphingomyelin because tannic acid binds to PC only at much higher concentrations than what is required to antagonize SMase C activity (13; 26). Furthermore, the EC50 (∼50 nM) of tannic acid for antagonizing the SMase C-catalyzed sphingomyelin hydrolysis in a biochemical reaction (involving no PC) is comparable to its EC50 (∼100 nM) for antagonizing SMase C suppression of CFTR activity in a biological membrane (Figure 3).”

Thus, the PC-PLC result is no longer necessary and we have now removed the PC-PLC data.